# Cancer Survivors in Saint Lucia Deeply Value Social Support: Considerations for Cancer Control in Under-Resourced Communities

**DOI:** 10.3390/ijerph19116531

**Published:** 2022-05-27

**Authors:** Aviane Auguste, Shania Cox, JoAnn S. Oliver, Dorothy Phillip, Owen Gabriel, James St. Catherine, Carlene Radix, Danièle Luce, Christine Barul

**Affiliations:** 1Vaughan Arthur Lewis Institute for Research and Innovation (VALIRI), Sir Arthur Lewis Community College, Morne Fortune, Castries LC06 101, Saint Lucia; stcatherine1950@gmail.com; 2Univ Rennes, Inserm, EHESP, IRSET (Institut de Recherche en Santé, Environnement et Travail)-UMR_S 1085, 97100 Pointe-à-Pitre, France; shaniabicar@hotmail.com (S.C.); daniele.luce@inserm.fr (D.L.); christine.barul@inserm.fr (C.B.); 3Faculté de Médecine, Campus de Fouillole, Université des Antilles, 97157 Pointe-à-Pitre, France; 4Capstone College of Nursing, The University of Alabama, Tuscaloosa, AL 35401, USA; joliver@ua.edu; 5Faces of Cancer Saint Lucia, Tapion Ridge, Castries LC04 201, Saint Lucia; facesofcancerslu@gmail.com; 6Department of Oncology, Owen King European Union Hospital, Millenium Highway, Castries LC04 201, Saint Lucia; gabrielowen2000@gmail.com; 7Caribbean Association for Oncology and Hematology, Belmont, Port of Spain 150123, Trinidad and Tobago; 8Organisation of Eastern Caribbean States (OECS) Commission-Franck Johnson Avenue, Morne Fortune, Castries LC06 101, Saint Lucia; carlene.radix@oecs.int

**Keywords:** cancer, community health, health disparities, social support, small island developing state, low and middle-income countries, Caribbean, Saint Lucia

## Abstract

Understanding the views of cancer survivors on their experience is important for informing community-based interventions. We studied, for the first time, the views of cancer survivors residing in Saint Lucia on their overall care experience. We used interview data from a cohort of adult cancer survivors from Saint Lucia between 2019 and 2020. We performed a thematic analysis to derive themes from codes. Forty-four survivors provided responses to at least one of the three questions. The majority of survivors were black, female and diagnosed with breast cancer. Survivors were interviewed on average five years after diagnosis. Four common themes emerged; “Availability of support groups”, “Importance of support from family and friends”, “Access to finances” and “Health education and patient navigation”. Travel overseas for health services was common among survivors. Survivors expressed emotional distress during travel due to isolation from family and local providers. This is typical among island populations and is distinct from existing patient frameworks. Survivors also suggested that networking amongst providers and interventions assisted families of cancer survivors. Although tertiary care services are limited, we showed that survivors deeply value and depend on their inter-personal relationships during care. Interventions aimed at strengthening the inter-personal environment of survivors are warranted.

## 1. Introduction

Persons living with cancer in under-resourced and vulnerable populations are disproportionately affected by their disease and have less access-to-care compared to persons from high-income settings [1]. In small island developing states, the challenges encountered when accessing care are further amplified. The health care systems in these islands are fragile and are constantly threatened by natural disasters. Implementation of innovation therapeutics and diagnostic services are difficult due to inherently small populations and limited economic capacity [2].

Saint Lucia is a small island developing state in the Caribbean. In 2020, there were 449 new cases of cancer and 232 deaths for a Afro-Caribbean population of about 180,000 [3]. There are three public hospitals, one private hospital, and many health centers distributed across the island. Universal health care does not exist in Saint Lucia. However, residents have access to a national social security system, which subsidises some health care costs and private health care insurance policies [4].

Different cancer advocacy groups operate on the island including “Faces of Cancer Saint Lucia”. Faces of Cancer Saint Lucia started in 2009 in order to assist patients going through their cancer journey. Faces of Cancer Saint Lucia has a membership of over 215 persons including volunteers from rural communities. All services and social events offered to survivors of the group are free of charge. Services include education, health fairs, emotional and spiritual support, and chemotherapy and post-surgical support [5]. Faces of Cancer Saint Lucia is currently developing activities beyond cancer advocacy towards involvement in patient navigation.

Having an in-depth comprehension of patients’ experiences during the delivery of cancer care is of growing value as it improves understanding of patient expectations, and therefore informs community interventions to accompany cancer survivors throughout their journey aiming for better survivorship [6].

We have knowledge of cancer care experience from pacific islanders [7] and other populations from small islands and under-resourced populations [8,9,10]. Survivors from these populations have expressed financial barriers to care, difficulties navigating the health system, and travelling long distances for care due to geographic isolation. These survivors rely heavily on assistance and encouragement from family, and members from the community and faith-based organizations as a means of psycho-social support. Patient navigation programs are also highly valued by these survivors, and have been used to improve equity with counterparts from high-resource settings [6,7].

The under-resourced populations from the Asian/pacific region and Africa for which data is available live either in large countries with developing economies or live on small islands governed by a developed country. However, unlike these populations, Caribbean populations are mostly small island developing states. This status often leads cancer survivors to travel overseas for care in the hope of accessing better services [11]. However, this practice could be counterproductive as it exposes survivors to isolation, which can worsen health outcomes [12,13].

Considering the socio-cultural differences, varying levels of access-to-care, and health seeking behaviors between regions (Asian/pacific, Caribbean and Africa), patient perceptions may not be the same in the Caribbean. Data on this in the Caribbean are scarce. A previous study quantitatively assessed the overall care experience [11]. The only qualitative study on perception of cancer health services was from a health care provider perspective of breast and cervical cancer patients [8].

We sought to describe for the first time the views of cancer survivors residing in Saint Lucia on their overall care experience using a qualitative method.

## 2. Materials and Methods

The present study is a secondary analysis drawing data from a community-based study referred to as “the DCAP study“(Description of the Cancer Health Services: Diagnosis and Treatment Pathways). The protocol for this study has been fully described elsewhere [11].

### 2.1. Patient Recruitment

The DCAP is a cohort of cancer survivors between May 2019 and August 2020. Eligible patients were greater than 18 years of age, able to communicate in English or Creole (without cognitive impairment), with an invasive cancer diagnosis (any cancer site, histology, and year of diagnosis), and having accessed health services in Saint Lucia due to cancer. Participation included authorization to access a patient’s data from medical records in health care institutions and centres. Sources for subject recruitment were Faces of Cancer Saint Lucia (FOCS), Victoria Hospital, the Oncology centre, and key informants. Patients at health care establishments were recruited during opportunistic cancer navigation assistance by a FOCS representative. Key informants were recruited using purposive sampling. We aimed to constitute a sample that would reflect the cancer survivors in Saint Lucia by sex, cancer site and district of residence. When possible, we recruited key informants during cancer advocacy activities organized by FOCS. Snowball sampling was used during interviews to identify prospective participants [14,15]. We screened data sources for potentially eligible participants and then invited as many patients as possible. Next-of-kin were interviewed where the index patient was deceased, or not physically able to undergo an interview.

The DCAP study was granted ethics approval by the ethics committee from the Medical and Dental Council (Saint Lucia, WI). All participants provided written informed consent prior to the study-required interview.

### 2.2. Data Collection and Questionnaire

Eligible participants were interviewed face-to-face by trained field investigators using a standardized questionnaire. The content of this questionnaire has been fully described elsewhere [11]. Participants were asked to have on-hand their test reports and personal clinical documents, to use as memory-aids during interviews.

The questionnaire was developed to ascertain sociodemographic variables such as education level, private medical insurance, hot water at home, employment and clinical characteristics, such as cancer stage at diagnosis, and comorbidities. Participants’ personal appraisal of their experiences for major events was ascertained throughout the interview. Interviews took place at the Faces of Cancer office or at the participants place of residence. They lasted on average one hour and 24 min (standard deviation: 34 min).

This current study was based on three open-ended questions ascertaining information on their overall care experience (clinical and nonclinical aspects). Cancer survivors were asked specifically “Was there anything in particular that made your experience easier?”, “Was there anything in particular that made your experience harder?”, and “Do you have any suggestions to help improve the experience for other people in similar circumstances?”. Probing was not used for these questions during interviews.

### 2.3. Variables and Definitions

Education level refers to the highest level of education that was completed. Private health insurance refers to coverage at the moment of the interview regardless of the person who pays the policy. Hot water at home refers to the availability of hot running water through a heating system in their primary place of residence. Professional status refers to a form of paid employment at the moment of the interview. The variable was divided into two categories: still working and not working. Not working includes unemployment, volunteer work and retirement. Diagnosis abroad was defined as a medical test performed that required physical travel outside of Saint Lucia. Treatment abroad was defined as a therapeutic intervention administered outside of Saint Lucia.

### 2.4. Data Analysis

We extracted the responses for the three questions based on patient overall care experience. Two of our authors independently analyzed and coded the same sample of responses. Interview responses were blinded for the other variables (age, sex, cancer site etc.) to prevent them from influencing the research findings.

After saturation was reached, a thematic analysis approach was used to categorize key codes into themes and subthemes. Thematic analysis is a rigorous, yet inductive, set of procedures designed to identify and examine themes from textual data in a way that is transparent and credible. This method draws from a broad range of several theoretical and methodological perspectives, but the aim is ultimately to present stories and experiences voiced by study participants as accurately and comprehensively as possible [16]. Guest et al. described basic steps in undertaking thematic analysis [16]; Familiarization with and organization of transcripts; Identification of possible themes; Review and analysis of themes to identify structures. Coders met at different intervals to discuss emerging themes until a general consensus was achieved.

## 3. Results

### 3.1. Characteristics of Cancer Survivors

Of the 50 cancer survivors from the initial DCAP study, 44 provided responses for this current analysis. Table 1 shows the sociodemographic and clinical characteristics of those participants. The majority of survivors were black, female, and diagnosed with breast cancer. On average, survivors were 53 years at diagnosis and were interviewed about five years after (standard deviation: 5.3). Years of survivorship were heterogeneous. Most participants were interviewed between two and eight years after their diagnosis. A little over a quarter of participants were more recently diagnosed (<2 years). At diagnosis, 62% reported an early-stage cancer and 47% reported a history of medical conditions. The most frequent conditions reported were hypertension (32%) and diabetes (14%). The majority (73%) had finished their initial active treatment at the time of their interview. In terms of socioeconomic variables, at least half of these survivors had a spouse, a professional activity and hot water at home. Only 40% of survivors were covered by private health insurance. Twenty-eight percent had only primary school education. More than half of the survivors had cancer treatment done outside of Saint Lucia. The proportion of survivors travelling for diagnostic tests was slightly higher (66%).

### 3.2. Thematic Analysis of Patient Experiences

We analyzed the responses from the three open-ended questions for which participants provided a response. Based on the responses, saturation was achieved. We noted 39 responses for the question “Was there anything in particular that made your experience easier?”, 37 for the question “Was there anything in particular that made your experience harder?”, and 41 for the question “Do you have any suggestions to help improve the experience for other people in similar circumstances?”. The length of responses were mostly one to three sentences. On one hand, family support was by far the most common code among the responses for the positive aspects of care, and represented about half of the survivors. On the other hand, responses for the negative aspects and suggestions were more heterogeneous. Fifteen themes emerged from the three open-ended questions (Figure 1). Table 2 shows key quotes from survivor responses contributing to the development of themes. Of the 44 interviews conducted, three were with caregivers/next-of-kins. Caregiver codes were similar to those from cancer survivors and did not contribute to any distinct themes (Appendix A). We identified four themes that were common to the three questions: (1) Availability of support groups, (2) Importance of support from family and friends, (3) Access to finances, and (4) Health education and Patient navigation. Hereafter, we provide a detailed analysis of the content leading to the formation of these themes.

#### 3.2.1. Availability of Support Groups

A total of 15 categories were identified to create this theme. Survivors expressed how support groups assisted them in getting information and in coping strategies with their illness, “*Faces of cancer because of information they gave and hope*”. Relating to a more difficult experience, many survivors expressed their dissatisfaction with not having the knowledge of existing support groups, “*Not having knowledge of who to contact for support*”. Survivors also gave advice to other cancer survivors relating to cancer groups. Many insisted that it is quite valuable throughout the journey to share your experiences with other survivors. “*We need a cancer center to provide support and*
*counselling to patients who are diagnosed with cancer. Increased support to Faces of Cancer to assist patients in care and treatment*.”

#### 3.2.2. Importance of Support from Family and Friends

Support from family and friends is one of the most frequent themes in this study. Survivors who had an easier journey expressed their gratitude for having family members and close friends supporting them throughout the journey, “*The only thing that made my experience easier is the fact that my sister accompanied me at every visit to the doctor and to the hospital*”. Survivors who travelled overseas for care spoke about the burden associated with being isolated from their family “*I could not see my husband and children*”, ”*Having to leave my son in Saint Lucia made it a bit hard*”.

#### 3.2.3. Access to Finances

Many survivors felt overwhelmed with the burden of finding the funds to pay for their treatment, laboratory tests, medication, etc. throughout their journey. Survivors showed their disappointment with the lack of enthusiasm from medical professionals to assist them when they were unable to make payments, “*Having no finance to pay for treatment and the doctors would not see you if you have no money, they would rather you die*”. Some survivors emphasized the need to have medical insurance early enough in the case of being diagnosed with cancer. A common way of paying for treatment as expressed by survivors was by raising funds whether it be in the form of having fundraiser barbeques created by family members, close friends or even members in the community, “*…Just if one doesn’t have insurance one should start asking for money early so they can do the treatment without missing any treatment*”. On the other hand, one survivor described their experience as being easier since he/she “*did not have to worry about finances*”.

#### 3.2.4. Health Education and Patient Navigation

Survivors have shown profound interest in having a navigation system within cancer care. Survivors are burdened with having no knowledge on “what’s next” after being diagnosed. They feel that inadequate information is provided by health care expertise after their diagnosis; “*Lack of team structure to deal with issues together…*”. Other survivors also expressed the need for health care professionals with the support from government to raise awareness, to educate the public, and to invest in cancer research. However, one survivor who had knowledge of the disease described having a more difficult experience, “*Having the knowledge and being the patient is heart wrenching*”.

## 4. Discussion

This is the first study focusing on cancer care experiences from a patient perspective in the Caribbean. This study underscores the importance of family and social support for a positive cancer experience in Saint Lucia.

Although the themes that emerged from our study were widely consistent with those from existing patient experience framework (patient preferences, emotional support, physical comfort, information and communication, continuity and transition, coordination of care, involvement of family and friends, and access-to-care) [17,18], we showed that survivors often have to travel overseas and leave their family in Saint Lucia to access care, and this impacted negatively on other aspects of the cancer experience. This finding is novel, and distinguishes our survivors from those in larger countries.

We compared our themes with the few data from other small islands. Themes from the Caribbean islands of Dominica, Grenada, and Saint Vincent and the Grenadines were consistent with ours [8]. Our survivors had a deep appreciation for the support from family and friends, and for support groups like Faces of Cancer Saint Lucia. Interestingly, the indigenous populations from the Torres strait islands of Australia also appear to have similar cancer experiences to our survivors [7]. They both have similar experiences while they access care overseas (e.g., isolation from family, language barrier, and cultural differences).

However, compared to these same islands, survivors from Saint Lucia had great enthusiasm for religion and faith, a factor which was uncommon in other small developing islands outside of the Caribbean [7]. Many survivors from our study looked towards higher spiritual powers, praying and having faith as a means of support or coping mechanism. The role of religion and spirituality has also been described in larger LMICs [10]. However, compared to developed countries, we believe that family support and faith appeared to be valued more among our survivors in Saint Lucia and other LMICs [19,20,21].

Access to finance was a recurring theme across the three open-ended questions. Cost of treatment and obligation to fundraise were mostly cited as sources of financial hardship. Financial hardship is well-known to be associated with the cancer experience in both high-income and low-income countries [22,23]. Qualitative studies on the financial hardship in LMICs and small islands are particularly scarce [23]. One study using a semi-structured interview was conducted in Iran [24]. Iranian survivors reported financial difficulties due to interference with their ability to work [24]. This was not observed in our study.

We hypothesized a possible relationship between some themes. Survivors expressed concerns with lack of empathy from providers, trust in their expertise, and health education. Lack of empathy may explain the distrust in expertise and motives of providers [25]. Consequently, the patients–provider relationship is suboptimal and diminishes the quality of knowledge transfer and care [26,27].

In addition, many survivors also spoke about emotional distress. Travelling to more developed countries for treatment often resulted in leaving behind family. Knowing the emphasis placed on family support by our participants, there is likely a strong patient-burden generated by travelling for care. This may contribute negatively to health outcomes of survivors from Saint Lucia. This link between social support and improved quality of life is well established [9,28,29]. A previous study on breast cancer patients showed that having social support mediates the choice of coping strategies toward positive reframing, which leads to better emotional well-being [28]. Social support is defined as a network of family, friends, neighbors, and community members that is available in times of need to give psychological, physical, and financial help [30].

Our findings add further understanding to what survivors in Saint Lucia view as important during their care. Patient satisfaction appears to be influenced more by inter-personal and provider factors rather than objective system measures. In a previous study, quantitative ratings of overall care experience showed 76% of these cancer survivors from Saint Lucia were satisfied (rated as “good/very good”) [11], whereas the responses of our current analysis were heterogeneous, revealing both a strong role of family/friend support and also numerous difficulties. Indeed, our current analysis revealed notable dissatisfaction with the medical professionals’ delivery of care, notably when conveying vital information and timely diagnosis. This incongruity between quantitative ratings and qualitative responses raises two potential explanations on the perception of the overall experience of cancer survivors in Saint Lucia. Firstly, we believe that the effect of psycho-social support may outweigh that of suboptimal health services in Saint Lucia. Secondly, health literacy is also a probable factor as we previously described [11]. Survivors may not view long delays or low-quality services as problematic due to lack of knowledge of best practices and standards in cancer care. There may be other factors that contribute to the perceptions of cancer survivors. This information is vital for tailoring interventions. Future work should particularly assess potential associations between the socioeconomic status and patient perceptions of care using a mixed methods approach [31].

This study had several strengths and limitations. Recall bias is likely since we ascertained information on patient experiences several years after their diagnosis. However, the effect of recall bias is unlikely to be alarming. The average delay was only five years, and our major themes corroborate with data from providers treating cancer patients from Saint Lucia [8]. Given the purposive sampling performed for this study, our results may not be an accurate representation of the views of all cancer survivors in Saint Lucia, a common weakness of qualitative studies. Some survivors were also recruited by Faces of Cancer Saint Lucia. Consequently, responses on support groups may have been overrepresented. In addition, most of the sample comprised women. Greater participation from women is a common occurrence in research studies [32]. However, we previously showed that our sample was indeed representative of the most common cancer sites by sex in Saint Lucia [11]. In addition, 44 participants spoke about the factors they perceived as important. This is a substantial sample size for a qualitative investigation, and is also comparable to other studies on this topic [7,9]. Furthermore, our study also adds new information to cancer care from a patient’s and caregiver’s perspective. The patient perspective gives a more holistic depiction of the difficulties encountered unlike studies from a provider perspective where comments on provider performance may be omitted [8]. In addition, our investigation included both quantitative and qualitative data.

Small developing islands like Saint Lucia are not always equipped with the resources needed to adequately treat cancer patients. The qualitative data gathered from this study raises awareness of the importance of capturing patients’ perspectives when receiving treatment. With our description of patients’ experiences, the local government and patient associations have an opportunity to plan and implement more successful evidence-based patient-centered interventions focusing on reducing the patient-burden associated with social isolation, notably from separation from family.

Although these patient experiences are from an island context, we showed that these findings are also relevant considerations for planning of cancer control in under-resourced and uninsured populations in more developed countries.

## 5. Conclusions

In light of our study, multi-disciplinary case-conferencing, patient education programs, and patient navigation could contribute to substantial improvement in the care experience and better survivorship among cancer survivors in Saint Lucia. These survivors depend highly on support from family for a positive cancer experience. Travelling overseas for more comprehensive care often translated into separation from family and an additional burden on survivors. The above interventions can be implemented immediately awaiting the development of more tertiary cancer services on-island. We believe that support groups such as Faces of Cancer Saint Lucia have an important role in building scientific evidence to strengthen the advocacy for better cancer control in Saint Lucia. Small developing islands of the Caribbean and similar under-resourced populations with limited offerings in cancer health services may also benefit from these new findings.

## Figures and Tables

**Figure 1 ijerph-19-06531-f001:**
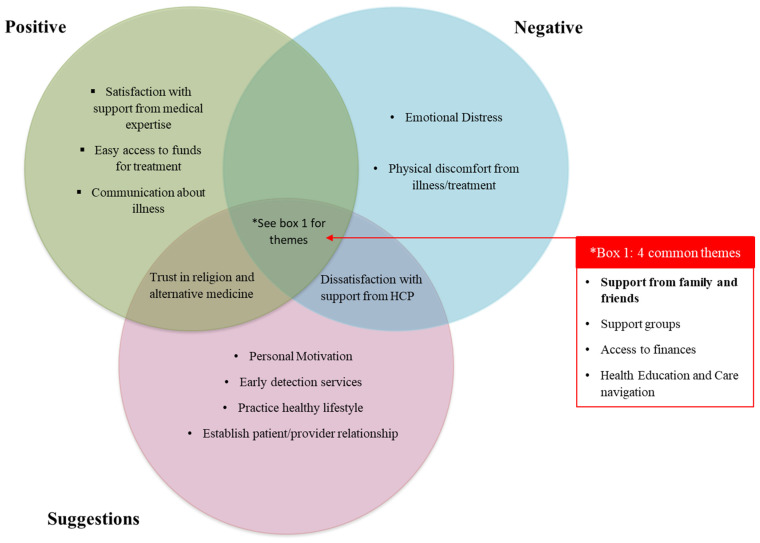
Venn diagram of significant themes emerging from analysis on the responses from the three open-ended questions. Saint Lucia (West Indies), 2019–2020. Each circle represents an open-ended question; the main themes that emerged are written within. The intersection between each circle represents the themes that were common to each question. Themes common to all three open-ended questions are represented in box 1. Positive: “Was there anything in particular that made your experience easier?”, Negative: “Was there anything in particular that made your experience harder?” and Suggestions: “Do you have any suggestions to help improve the experience for other people in similar circumstances?”. HCP: Health care provider.

**Table 1 ijerph-19-06531-t001:** Sociodemographic characteristics of the 44 cancer survivors included in the current study who had responses for at least one of the three questions.

Characteristics		
**Sex, *n*%**		
Men	10	22.7
Women	34	77.3
**Age at diagnosis (y** **)**		
Median (IQR)	55	(42–64)
**Cancer site, *n*%**		
Breast	25	56.8
Female pelvis ^a^	8	18.2
Prostate	6	13.6
Other ^b^	5	11.4
**Survivorship (y** **)**		
Median (IQR)	4.2	(1.7–7.8)
Missing	1	
**Stage at diagnosis, *n*%**		
Early (I/II)	24	61.5
Advanced (III/IV)	15	38.5
Missing	5	
**Treatment status, *n*%**		
Finished initial active treatment	32	72.7
Still on treatment	8	18.2
No treatment taken	4	9.1
**Marital status, *n*%**		
Single	19	44.2
Married/Other	24	55.8
Missing	1	
**Education level, *n*%**		
Primary	12	27.9
Secondary	13	30.2
Tertiary	18	41.9
Missing	1	
**Private medical insurance, *n*%**		
Yes	18	40.9
No	26	59.1
**Hot water at home, *n*%**		
Solar	12	27.9
Electric	9	20.9
No	22	51.2
Missing	1	
**History of medical condition(s), *n*%**		
Yes	21	47.7
No	23	52.3
**Professional status, *n*%**		
Working	24	55.8
Not working	19	44.2
Missing	1	
**Treatment abroad, *n*%**		
Yes	22	57.9
No	16	42.1
Missing	6	
**Diagnostic test(s) abroad, *n*%**		
Yes	27	65.9
No	14	34.2
Missing	3	

Saint Lucia (West Indies), 2019–2020. IQR: Interquartile range. ^a^: Cervix *n* = 3, Endometrium *n* = 5, Ovary *n* = 2. ^b^: Colon (3 men), Parotid gland (1 woman) and Leukaemia (1 man).

**Table 2 ijerph-19-06531-t002:** Key quotes from participant responses contributing to the development of themes.

Open-Ended Question	Patient n°	Quotes
Was there anything in particular that made your experience easier?	1	*Joining Faces of Cancer Saint Lucia.*
	2	*The support of family and friends who provided housing, spiritual and emotional support.*
	3	*Yes, the almighty, I trusted him to give me the strength to endure.*
	4	*Family support (My sister was always here), insurance (Money was not a problem), my employer supported me mentally and financially.*
	5	*Treatment at Tapion hospital was excellent but costly.*
	6	*Family support, natural medications.*
Was there anything in particular that made your experience harder?	7	*Coming up with the funds, some health care providers did not give me the opportunity to share information, they don’t listen. Emotionally I could not deal with the first diagnosis. I am still paying the loans.*
	8	*Dealing with cancer. Having no money or place to turn to. No support group available.*
	9	*Lack of team structure to deal with issues together. Absence of counsellors. Being discharged prematurely.*
	10	*Having to leave my son in Saint Lucia made it a bit hard. My son was afraid of me.* *I had to send him back to Saint Lucia.*
	11	*Having no finance to pay for treatment and the doctors would not see you if you have no money, they would rather you die.*
	12	*Not knowing where to go to get help in dealing with the illness.*
Do you have any suggestions to help improve the experience for other people in similar circumstances?	13	*Advising on early detection exam for all types of cancer. Visits to health centres, health promotion, seek support and other medical interventions including cancer markers.*
	14	*Having a support group with cancer patients. Having a psycho-social person attached to oncologist or hospital pre/post diagnosis. Something financial in place for persons living with cancer. Government should invest in funding cancer research and treatment because of its cost. Home care for the patients with cancer. Build a wing at the hospital just like the maternity to deal with cancer patients. Decentralise train persons in palliative care.*
	9	*Doctors need to come together as a team (GP, surgeon, oncologist, radiologists). Everything is done in isolation because they all want to make a fortune. The medication prescribed was not available locally and is $100 US monthly. People don’t understand the importance of health insurance. Inculcate healthy lifestyle in persons. Sensitize and educate the public. Availability of treatment needed. Never give up. Come out and let people or family know of your condition. Live a stress-free life (reduce stress level). Carers of persons with cancer ensure that they are given care. Have to support you at all times. Do your breast examination. Know your family medical history.*
	15	*I would like to encourage persons in similar circumstances to keep the faith and continue praying.*
	16	*Do whatever you need to raise funds for treatment. Listen to the health care providers and do not waste time.* *Do regular cancer screening. Continue or adapt healthy life styles.*

Saint Lucia (West Indies), 2019–2020.

## Data Availability

The datasets analyzed during the current study are available from the corresponding author on reasonable request.

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
