# Peer review of "Cancer Survivors in Saint Lucia Deeply Value Social Support: Considerations for Cancer Control in Under-Resourced Communities"

_ijerph, 2022, doi:10.3390/ijerph19116531_

Round 1

Reviewer 1 Report

I believe the manuscript has been significantly improved and now warrants publication in ijerph.

Reviewer 2 Report

I very liked the manuscript of the Authors, since I think they have chosen a good way to conduct this study:  it's very important understand more deaply different point of view of care.
There are only few concern to address:

  • insert number of Ethical approaval
  • check table 1 (eg in text, Authors say "On average, survivors were 53 years at diagnosis", instead in the table are indicated 55)
  • I suggest to little expand the description about Thematic Analysis by 
    Guest, MacQueen and Namey, 2012, for to be easily understood by both expert readers and newcomers  to the sector.

Reviewer 3 Report

The authors approach a very sensitive topic for the health systems all-over the world, but it seems more important in Saint Lucia. Many cancer survivors need help and this study can be the beginning of some good projects. Overall, the paper is well written, although I have some remarks to make, some suggestions that I think can help readers to have a clearer understanding of the information.

Reference [3] is from 2018, but the information in the lines 48-49 is specified to be from 2020.

I understand the study is about cancer survivors but the authors interviewed next-of-kin in case of deceased patients (lines 108-109)

Lines 163-164: at least half of these survivors had a spouse, a professional activity, private health insurance and hot water at home, but in the table it is written that only 40.9 had private insurance.

Line 95: May 2019 and August 2020, but line 191: Saint Lucia (West Indies), 2019-2021.

Regarding family support as a difficulty- not telling the family or not receiving support from the family even if they knew? If not telling- it can be proposed as a suggestion, if not receiving or not having a family close to them it can be a difficulty. But I think is better not to mix them.

I understand that the study is about cancer survivors, but still in some cases it was a family member or another person who took part in the study. Given that this is a qualitative study about the personal feelings/opinions of the patients (cancer survivors), in my opinion it is not suitable to mix the views. For example, is not the same the difficulty encountered by a family member when taking care of the patient with the difficulty encountered by the patient him/herself.

I think it would be useful for the quotations to have an identifier. At least like “participant 1/2/3/etc.” In Table 2 there is a mixture of quotations from various participants, without being specified each quotation belonged to which participant (as a number indicated for each participant at least).

In the Discussion section, the authors do not discuss the main results/themes, but it seems they choose some random results. For example, the authors discuss as a relevant result the spiritual/religious aspects, but in the results section they do not give too much relevance to this result. Similarly, about dissatisfaction with the professionals delivery of care. Instead, the authors do not discuss the financial theme- as a major common theme revealed by the analysis.

Lines 92-93: “The protocol for this study has been fully described elsewhere [11]. Lines 116-117: The content of this questionnaire has been fully described elsewhere [11]” From these words I understand this study includes the same participants as the study cited [11]. But in the same time, in lines 293-297 you treat them as separate studies, with separate participants: “Our findings add further understanding to what survivors in Saint Lucia view as important during their care. Patient satisfaction appears to be influenced more by inter-personal and provider factors rather than objective system measures. In a previous study, quantitative ratings of overall care experience showed 76% of these cancer survivors from Saint Lucia were satisfied [11].” Is it possible to have different results from the same participants in the same study? I think something is missing here.

Line 190: Venn diagram of significant themes, but only one of the four common themes is shown. So I do not necessarily see the relevance of the Figure 1, where only one common theme is shown.

Do you use social support in the discussion section as a synonym for support groups and support from family and friends? I see a lot of discussion on this social support, but it is not very clear exposed.

Author Response

This manuscript is a resubmission of an earlier submission. The following is a list of the peer review reports and author responses from that submission.

Round 1

Reviewer 1 Report

This manuscript explores qualitative responses to three survey questions from interviews with cancer survivors in Saint Lucia. The potential of exploring cancer experiences in an under-studied context is intriguing; however, the study was not designed to examine these potentially unique contextual influences, and, as a result, the findings do not contribute much new knowledge to the literature. Specific comments are noted below:

  • Introduction: A stronger case should be made for why this study is needed. It is insufficient to say that few studies have been conducted (though this is important). For example, explain more why findings from other places might be different in this context, what kinds of information are needed, and why choose a qualitative approach for this particular study. A little more detail is also needed about what we already know from previous studies – did the quantitative study noted use data from the same survey used in this study? And the last few paragraphs need more citations to support statements made.
  • Page 2, 2nd paragraph: Please add a citation and/or website for Faces of Cancer Saint Lucia so readers can learn more.
  • Methods: Please clarify the sampling approach (convenience + snowball sampling?). How were people identified? Was it convenience? It sounds later like all DCAP patients were approached? It’s also not clear in terms of timing if people were initially approached for this study in the methods described or later in another way after they participated in the DCAP study. Convenience sampling should be mentioned as a study limitation.
  • The term “study-required interview” is concerning. It sounds like people had to do the interview and did not have the option to disenroll from the study if they didn’t want to do the interview, though I assume this was not the case.
  • More details are needed about the interviews: How long did they take to complete? Where were they conducted? Did the interviewers using probing to expand upon and clarify participants’ responses to the open-ended questions?
  • Data analysis: Why were interview responses blinded for other variables (age, sex, cancer site etc.)? It seems like these could be important factors that could have been examined during the data collection and analyses. Also, it is said that a grounded theory approach was applied to the coding, but then a different method by Guest is mentioned. Grounded theory typically leads to the development of theory, which is not presented.
  • A main concern is the nature of the data. For example, did most responses to the 3 questions consist of paragraph-length responses plus responses to probes (at one end of depth) or were most responses 1-word answers (at the other end of the spectrum)? It’s not clear that the data were rich enough to really understand participants’ experiences from their perspectives, especially since the 3 questions were pretty vague. Relatedly, it’s not clear that saturation was achieved given the range of topics mentioned by participants.
  • Tables: The centered text is difficult to read – please left-justify.
  • Table 1: Some of the data are confusing as currently presented – e.g., it appears to be an age range in the “%” column. As IQR is noted, the % or other indicators of data type could similarly be placed in the rows instead.
  • Section 3.2: What does “for which patients were not indifferent” mean”?
  • Discussion and Conclusion: The goal of the study seems to be that there are unique aspects of experiencing cancer in Saint Lucia that need to be better understood. However, the interview questions were not designed to explore that these differences may be. Thus, although some of the findings may be different from other studies in island nations, the findings on the whole are not very surprising – many of these themes, like the role of religion and spirituality, have been well-documented in global research. It’s not clear that these findings offer something substantially new.
  • There is an overall need for grammatical editing re: commas and hyphens.

Reviewer 2 Report

This paper reports the results of a grounded theory analysis of responses to three open-ended questions from interviews of a cohort of 44 adult cancer survivors residing in Saint Lucia between 2019 and 2020. Overall, the paper is reasonably well written and the qualitative analysis and findings are well described and well done.

Here is one item to attend to in a revision. In Section 2.4, the text states: "Guest et al described four basic steps in undertaking thematic analysis [11]; Familiarization with and organization of transcripts; Identification of possible themes; Review and analysis of themes to identify structures." The sentence indicates four basic steps, but the following phrase appears to identify only three steps.

Reviewer 3 Report

The study “cancer survivors in Saint Lucia”, describing the experience of cancer survivors follows a qualitative approach that is certainly well suited as a first step towards a better understanding of the situation in a small developing state. The study is carefully conducted and well described. I have only a few major points, which are listed below in the order they appear in the manuscript.  

Introduction:

  • At the end of the introduction, explicitly state the question without already interpreting it.

Results:

  • How should Figure 1 be understood? How should the four general themes be read out here?

Methods:

  • The three open questions are very general. In a semi-structured interview, these could be introductory questions that are then used to go into more depth.
  • Please list the three open-ended questions in the methods section exactly as they were asked.
  • Please go into more detail about the fact that the diagnosis was made 5 years ago on average, also in the methods section. How was the recruitment planned exactly?

Discussion:

  • The results are not sufficiently placed in a larger context. They don't seem surprising in themselves: in virtually all studies where cancer patients are asked what is important to them, it comes out that social support is hugely important. And that patients turn to their faith is not surprising in people who are church-affiliated, for example. Here, reviews would be helpful to be able to really classify and compare the results well.
  • Some of the discussion raises issues that are coming up here for the first time, such as patients having to go abroad and leave their family at home.
  • The sample includes mainly women, this should be discussed. I.e., the discussion should pick up on this fact and include international literature here as well.
